# Beta-Aminoisobutyric Acid Inhibits Hypothalamic Inflammation by Reversing Microglia Activation

**DOI:** 10.3390/cells8121609

**Published:** 2019-12-11

**Authors:** Byong Seo Park, Thai Hien Tu, Hannah Lee, Da Yeon Jeong, Sunggu Yang, Byung Ju Lee, Jae Geun Kim

**Affiliations:** 1Division of Life Sciences, College of Life Sciences and Bioengineering, Incheon National University, Incheon 406-772, Korea; bbs0808@naver.com (B.S.P.); thaihientu@gmail.com (T.H.T.); 8973215@hanmail.net (H.L.); lov2fiction@naver.com (D.Y.J.); 2Department of Biological Sciences, University of Ulsan, Ulsan 680-749, Korea; 3Department of Nano-Bioengineering, Incheon National University, Incheon 406-772, Korea; abiyang9@gmail.com; 4Institute for New Drug Development, Division of Life Sciences, Incheon National University, Incheon 406-772, Korea

**Keywords:** beta-aminoisobutyric acid, myokine, hypothalamus, inflammation, microglia, obesity

## Abstract

Beta-aminoisobutyric acid (BAIBA), a natural thymine catabolite, is involved in the beneficial effects of exercise on metabolic disorders. In particular, it has been reported to reverse the inflammatory processes observed in the peripheral organs of animal models of obesity. Therefore, this study aimed to investigate whether BAIBA improves hypothalamic inflammation, which is also tightly coupled with the development of obesity. We observed that treatment with BAIBA effectively reversed palmitic acid-induced hypothalamic inflammation and microglial activation in vivo. Consistent with these findings, we confirmed that BAIBA reversed body weight gain and increased adiposity observed in mice fed with a high-fat diet. Collectively, the current findings evidence the beneficial impacts of BAIBA on the imbalance of energy metabolism linked to hypothalamic inflammation.

## 1. Introduction

The hypothalamic neuronal circuit dynamically participates in the regulation of the energy metabolism of the whole body [1,2]. Recently, it has been well established that hypothalamic inflammation is a major clue accounting for the development of multiple metabolic disorders [3,4]. The hypothalamic inflammation coupled to metabolic enrichment leads to gliosis, which results in the proliferation and morphological changes of glial cells, including astrocytes, microglia, and oligodendrocytes [4,5]. Among them, microglia, which are the resident immune cells in the central nervous system (CNS), are one of the crucial cellular contributors that drive physiological or pathological processes associated with whole-body energy metabolism [6,7]. In particular, it is currently accepted that the activation of microglia in the hypothalamus during conditions of over-nutrition could be a critical pathogenic component in the initiation of obesity development [5,6,8,9]. Thus, it is noteworthy to identify strategies to overcome metabolic diseases by alleviating the hypothalamic inflammation associated with the microglia.

A growing body of evidence has demonstrated that exercise is an effective therapeutic intervention for treating obesity and the related secondary complications [10,11]. Intriguingly, voluntary exercise gives rise to anti-inflammatory effects against fat-rich diet-induced hypothalamic inflammation [10,12]. However, the endocrine substances that mediate the benefits of exercise are yet to be elucidated. Recently, beta-aminoisobutyric acid (BAIBA) has been identified as a myokine originating from the skeletal muscles during physical activity. It has been reported to give rise to exercise-triggered protective effects against the development of obesity by regulating thermogenic programs in white adipose tissues in association with the proliferator-activated receptor-gamma coactivator-1α (PGC-1α)-dependent pathway [13,14]. Furthermore, a previous study has reported that BAIBA attenuates inflammation responses and insulin resistance in high-fat diet (HFD)-fed mice, and that this effect was reversed via the suppression of AMPK [15]. Based on these evidences, we designed the current study to explore the reversible effects of BAIBA on hypothalamic inflammation observed following treatment with a saturated fatty acid in vitro and long-term exposure to fat-rich diet in vivo. Furthermore, we investigated whether BAIBA elicits anti-obesity effects following the exposure of mice to a fat-rich diet.

## 2. Materials and Methods

### 2.1. Animals

Seven-week-old C57B/L6 male mice with an initial body weight of 20 ± 2 g (Dae Han Bio Link, Eumseong, Korea) were maintained under specific pathogen-free conditions at 22 °C and given access to food and water *ad libitum*.

Experiment 1: To examine the effects of BAIBA during the late stage of the development of obesity, the mice were subjected to acclimatization for a week, randomly divided into three groups, and fed either a standard diet (STD, 10% calories from fat, Research Diets Inc., New Brunswick, NJ, USA) or a high-fat diet (HFD, 60% of calories from fat, Research Diets Inc.) for 20 weeks. For BAIBA treatment, the mice were administered with normal drinking water (control group) or drinking water containing 150 mg/kg/day of BAIBA for 8 weeks (Scheme 1A).

Experiment 2: To further test the effects of BAIBA during the early stage of obesity development, the mice were fed either a standard diet or a high-fat diet for 8 weeks. Four weeks after the HFD treatment, the mice were administered with BAIBA (150 mg/kg/day of BAIBA dissolved in drinking water), while being continuously fed with the HFD (Scheme 1B). All the animal care and experimental procedures were performed in accordance with the protocols approved by the Institutional Animal Care and Use Committee (IACUC) at the Incheon National University (permission number: INU-ANIM-2019-05).

### 2.2. Intracerebroventricular (ICV) Cannulation

The mice were anesthetized with an intraperitoneal (i.p.) injection of tribromoethanol (250 mg/kg, Sigma-Aldrich, St. Louis, MO, USA) and placed in a stereotaxic apparatus (Stoelting, Wood Dale, IL, USA). The cannula (26 gauge) was implanted into the right lateral ventricle (coordinates: 0.1 mm lateral, 0.03 mm posterior, and 2.4 mm ventral to the bregma) of the mice and secured to their skulls using dental cement. The animals were kept warm until they recovered from the anesthesia and then were placed in individual cages. After surgery, a recovery period of 7 days was allowed before starting the experiments. To evaluate the effects of BAIBA on saturated fatty acid-induced microglial activation in the hypothalamus, palmitic acid (25 pmol/µL) was injected through the cannula after the systematic pre-treatment of the mice with BAIBA (3 mg/kg; i.p. injection).

### 2.3. Immunohistochemistry

The mice were anesthetized and perfused transcardially with 0.9% saline (*w*/*v*), followed by fixation of the hypothalamic tissues with 4% paraformaldehyde in phosphate buffer (PB; 0.1 M; pH 7.4). The brains of the mice were isolated and fixed overnight with 4% paraformaldehyde in PB. The coronal sections (thickness, 50 μm) were prepared using a vibratome (5100 mz Campden Instruments, Leicestershire, UK). Next, the sections were washed several times with PB, and preincubated with 0.3% Triton X-100 (Sigma-Aldrich) for 30 min at room temperature (RT). Then, they were incubated with the primary antibodies (antibodies against rabbit Iba-1, 1:1000 dilution, Wako, Osaka, Japan; rabbit cyclooxygenase-2 (COX-2), 1:1000 dilution, Abcam, Cambridge, UK) overnight at RT. Immunofluorescence was performed with the secondary antibodies (Alexa Fluor 594-labeled anti-rabbit antibodies, 1:1000 dilution, Invitrogen Life Technologies, Carlsbad, CA, USA) for 2 h at RT. For diaminobenzidine (DAB)-based IHC analysis to detect Iba-1 and COX-2, the sections were washed thoroughly and incubated with biotinylated anti-rabbit secondary antibodies, Avidin-Biotin Complex (ABC) reagent (Vector Laboratories, Burlingame, CA, USA), and the DAB substrate (Vector Laboratories). The sections were then mounted onto glass slides and coverslips were then placed on these slides with a drop of mounting medium (Dako North America Inc., Carpinteria, CA, USA). The coverslips were sealed with nail polish to prevent the drying and movement of the samples under the microscope.

### 2.4. IHC Image Capture and Analyses

The images were recorded by fluorescence microscopy (Axioplan2 Imaging, Carl Zeiss Microimaging Inc., Thornwood, NY, USA). The sections containing hypothalamic nuclei (stereotaxic coordinates: between −1.46 and −1.82 mm from the bregma) were matched with the mouse brain atlas book (Paxinos and Franklin, 2001, the Mouse Brain in Stereotaxic Coordinates—second edition, San Diego, CA, USA, Academic Press) and subjected to the IHC analyses. Both sides of the bilateral brain region were analyzed (two brain sections per mouse). The number of immunostained cells and the intensity of immunoreactive signals in the microglia were measured manually using ImageJ v. 1.47 software (National Institutes of Health, Bethesda, MD; http://rsbweb.nih.gov/ij/) by an unbiased observer. All morphometric analyses were performed without prior knowledge of the experimental group from which the sections were obtained (blind).

### 2.5. Culture and Treatment of the Cells

The murine BV-2 microglial cells were cultured in Dulbecco’s modified Eagle medium (DMEM) containing high glucose (Gibco BRL, Grand Island, NY, USA) and 5% (*v*/*v*) fetal bovine serum (Gibco BRL) at 37 °C in humidified 5% CO_2_ conditions. For the gene expression assay, the cells were seeded at a density of 1 × 10^6^ cells/well in 6-well plates. After 24 h, the attached cells were pre-treated with 100 µM BAIBA for 16 h, followed by the treatment with 200 µM palmitic acid (Sigma-Aldrich) for 4 h. The palmitic acid was dissolved in ethanol and conjugated with 10% bovine serum albumin (BSA).

### 2.6. Primary Astrocyte Culture

Following decapitation of five-day-old C57BL/6 mice, the diencephalon was removed under sterile conditions and triturated in Dulbecco’s modified Eagle′s medium (DMEM) F-12 containing 1% penicillin-streptomycin. The cell suspension was filtered through a 100 μm sterile cell strainer to remove debris and fibrous layers. The suspension was centrifuged, and the pellet was resuspended in DMEM F-12, containing 10% fetal bovine serum (FBS) and 1% penicillin-streptomycin. The cells were then grown in this culture medium in 75 cm^3^ culture flasks at 37 °C and 5% CO_2_. When the cells reached confluence (at approximately nine days), microglia were separated from adhered astrocytes by shaking the culture at approximately 250 rpm for 2 h. The cells were then seeded onto 12-well tissue culture plates, previously coated with poly-d-lysine hydrobromide (50 μg/mL), after which, they were distributed at 7.5 × 10^4^ cells/well and incubated at 37 °C with 5% CO_2_.

### 2.7. Quantitative Real-Time PCR (qRT-PCR)

Hypothalami were dissected laterally 2 mm either side of the third ventricle from the optic chiasm to the posterior border of the mammillary bodies, and the thalamus dorsally. Total RNA from the hypothalamic tissues of the mice and cultured BV-2 cells was isolated and reverse-transcribed to yield cDNA using the Maxime RT PreMix kit (Intron Biotechnology, Seoul, Korea). Real-time PCR amplification of the cDNA was performed using the SYBR Green Real-time PCR Master Mix (Toyobo Co., Ltd., Osaka, Japan) in a Bio-Rad CFX 96 Real-Time Detection System (Bio-Rad Laboratories, Hercules, CA, USA). The results were analyzed using the CFX Manager software and the expression levels of the genes were normalized to the expression levels of the housekeeping gene *β-actin*. The primer sequences used were: *IL-1β*, F-AGGGCTGCTTCCAAACCTTTGAC and R-ATACTGCCTGCCTGAAGCTCTTGT; *IL-6*, F-CCACTTCACAAGTCGGAGGCTTA and R-GCAAGTGCATCATCGTTGTTCATAC; *TNF-α*, F-TGGGACAGTGACCTGGACTGT and R-TTCGGAAAGCCCATTTGAGT; *Iba-1*, F-AGCTTTTGGACTGCTGAAGG and R-TTTGGACGGCAGATCCTCATC; *CD11b*, F-CCACTCATTGTGGGCAGCTC and R-GGGCAGCTTCATTCATCATGTC, and *β-actin*, F-TGGAATCCTGTGGCATCCATGAAAC and R-TAAAACGCAGCTCAGTAACAGTAACAGTCCG.

### 2.8. Measurement of Cytokine Levels

The supernatants from cultured BV-2 cells were collected after BAIBA pre-treatment and the subsequent administration of palmitic acid (200 μM) for 24 h. The concentration of the released cytokines in the plasma was measured using a mouse IL-1β and IL-6 ELISA kit (R&D Systems, Minneapolis, MN, USA) according to the manufacturer’s instructions.

### 2.9. Measurement of Oxygen Consumption (VO_2_), Carbon Dioxide Production (VCO_2_), and Energy Expenditure

A standard 12 h light/dark cycle was maintained throughout the indirect calorimetry studies. Mice were acclimated to the metabolic cages for 24 h before data collection by using an indirect calorimetry system (Promethion, Sable Systems, North Las Vegas, NV, USA). Food and water were provided *ad libitum* and data were collected for three days after the acclimation. Oxygen consumption (VO_2_) and carbon dioxide production (VCO_2_) were measured for each mouse at 10 min intervals. Incurrent air reference values were determined after the analysis of the mice in all the cages. The respiratory quotient (RQ) was calculated as the ratio of CO_2_ production over O_2_ consumption. Data acquisition and instrument control were performed using the MetaScreen v. 1.6.2 software, and the obtained raw data were processed using ExpeData v. 1.4.3 (Sable Systems) with an analysis script detailing all the aspects of data transformation.

### 2.10. Statistical Analysis

Statistical analyses were performed using the Prism 6.0 software (GraphPad Software, San Diego, CA, USA). All the data are expressed as the means ± SEMs. Statistical significance was determined using the two-tailed Student’s *t*-test and one-way analysis of variance (ANOVA), followed by Sidak’s post-hoc tests. *p* values < 0.05 were considered statistically significant.

## 3. Results

### 3.1. Beta-Aminoisobutyric Acid (BAIBA) Improved Hypothalamic Inflammation in Mice with Severe Obesity Following the Long-Term Consumption of a Fat-Rich Diet

To confirm whether BAIBA has beneficial effects against the hypothalamic inflammation and the metabolic disturbance observed during conditions of chronic over-nutrition, we treated obese mice that were fed a fat-rich diet for 20 weeks with BAIBA for 8 weeks. As shown in Figure 1A, the long-term administration of BAIBA did not alter the body weight, epididymal fat weight, and liver weight in mice fed with the HFD (Figure 1A–C). However, we found that BAIBA treatment effectively reversed the increase in the plasma levels of IL-1β and IL-6 (Figure 1D,E), the pro-inflammatory cytokines, indicating that BAIBA had a reversible effect on inflammatory responses resulting from the long-term consumption of a fat-rich diet. Since hypothalamic inflammation is directly associated with the pathogenesis of obesity [16], we further ascertained whether BAIBA has an impact on the hypothalamic inflammation by evaluating the hypothalamic expression of genes that are involved in the inflammatory responses. The BAIBA treatment significantly ameliorated the increase in the levels of pro-inflammatory genes, including *TNF-α, IL1-β,* and *IL-6* (Figure 1F–H), in the hypothalamus of the HFD-fed mice, compared to the case for the hypothalamus of the STD-fed mice. These findings indicate that BAIBA elicits a beneficial effect on hypothalamic inflammation, acting against late-stage obesity pathogenesis without a reversible impact on obesity phenotypes.

### 3.2. BAIBA Reversed the Microglial Activation in Mice with Severe Obesity Following the Long-Term Consumption of a Fat-Rich Diet

During the last decade, it has been well established that the microglia in the hypothalamus dynamically participates in the development of metabolic abnormalities. In particular, the microgliosis accompanied by innate immune responses is tightly coupled to the hypothalamic inflammation in response to over-nutrition [3,17]. Thus, we further investigated the effect of BAIBA on the microglial activation observed in mice with HFD-induced obesity by immunohistochemistry analysis using an antibody against the Iba-1 protein, which is a molecular marker of microglia. Consistent with previous findings that have demonstrated the occurrence of microgliosis in the hypothalamus of HFD-fed mice with obesity, we found an increased number of microglial cells and an elevated concentration of the Iba-1 protein in microglia of the hypothalamic nuclei, including arcuate nucleus (ARC), ventromedial nucleus of the hypothalamus (VMH), and dorsomedial nucleus of the hypothalamus (DMH). BAIBA treatment successfully increased microglial activation in the hypothalamic nuclei, which are directly associated with the central control of energy metabolism (Figure 2A–C). Consistent with these histological findings, we observed that the enhancement of the mRNA levels of *Iba-1* and *CD11b* seen in the obese mice was significantly reversed by the BAIBA treatment (Figure 2D,E). Furthermore, BAIBA treatment significantly reduced the number of cells that synthesize cyclooxygenase-2 (COX-2) protein, a key enzyme in the production of prostaglandins, which also participate in cellular inflammatory responses (Figure 2F,G). These observations indicated that BAIBA ameliorated the hypothalamic inflammation by alleviating the microglial inflammation. 

### 3.3. BAIBA Ameliorated the Palmitic Acid-Induced Inflammatory Responses in BV-2 Microglial Cells

An elevation of the levels of circulating free fatty acids drives hypothalamic inflammation by enhancing the microglial activation [5]. Therefore, to verify the anti-inflammatory role of BAIBA in elevating the microglial inflammation induced by over-nutrition, the BV-2 cells were first treated with palmitic acid, a saturated free-fatty acid, and then with BAIBA. In accordance with previous reports, the palmitic acid treatment led to an increase in the mRNA levels of inflammatory cytokines such as TNF-α, IL-1β, and IL-6 (Figure 3A–C) in the cultured BV-2 microglial cells. This palmitic acid-induced elevation of the inflammatory responses was almost completely ameliorated by the exogenous BAIBA treatment (Figure 3A–C). In addition, treatment with BAIBA effectively rescued the palmitic acid-induced elevation of the expression of genes linked to microglial activation such as *CD11b* and *Iba-1* (Figure 3D,E). To further confirm the anti-inflammatory effects of BAIBA on palmitic acid-induced microglia inflammation, we treated hypothalamic primary microglia with BAIBA, 1 h after palmitic acid treatment, and then measured mRNA expression involved in microglia inflammation. The administration of BAIBA effectively moderated the increased mRNA expression, such as *Iba-1*, *TNF-α*, and *Cox-2* (Figure 3F–H), triggered by the pre-treatment of palmitic acid.

### 3.4. Systemic Administration of BAIBA Ameliorated the Palmitic Acid-Induced Microglial Activation in the Hypothalamus

The reversible effects of BAIBA on the palmitic acid-induced inflammatory responses in microglial cells were further confirmed. Hypothalamic samples from the mice that received ICV injections of palmitic acid after they were systemically administered with BAIBA were obtained these samples were subjected to IHC analysis using an antibody against the Iba-1 protein. We first found that the central administration of palmitic acid resulted in an increase in the number of microglial cells, as detected by the Iba-1 protein, and the intensity of the immunoreactive signals of Iba-1 in the hypothalamic samples (Figure 4A–C). Consistent with the findings of the cell-based experiments, which showed the anti-inflammatory effects of BAIBA on palmitic acid-induced microglial inflammation, we observed that the systemic administration of BAIBA effectively reversed the palmitic acid-induced microglial activation in the hypothalamus samples (Figure 4A–C). Collectively, these findings, together with the data shown in Figure 3, evidence that BAIBA has beneficial impacts on the hypothalamic inflammation associated with microglial activation under energy enrichment conditions.

### 3.5. BAIBA Reversed Obesity Phenotypes and Hypothalamic Inflammation during the Early Stage of Obesity Development

Based on our findings that showed that BAIBA has reversible effects on hypothalamic inflammation without any changes of the metabolic phenotypes in mice with severe obesity, we evaluated the effects of BAIBA treatment during the early stage of HFD-induced obesity. Although we did not find any changes of the metabolic phenotypes in response to BAIBA treatment under conditions of severe obesity, we confirmed that during the early stage of HFD consumption, the BAIBA treatment effectively reversed the obesity phenotypes, including body weight gain and increased adiposity (Figure 5A,B). In addition, we observed that the BAIBA treatment led to elevated oxygen consumption, carbon dioxide production, and energy expenditure (Figure 5D–F). Consistent with the data showing the anti-inflammatory effects of BAIBA and the BAIBA-mediated amelioration of microglial activation in the hypothalamus during the chronic over-nutrition period, the BAIBA treatment alleviated the hypothalamic microglial activation resulting from the short-term consumption of HFD (Figure 5G–I). Collectively, these data and the data shown in Figure 1 suggested that BAIBA treatment could be utilized as a therapeutic strategy for preventing obesity, or may be more effective for patients showing the properties of the early stage of obesity.

## 4. Discussion

The findings of the current study highlighted that BAIBA, a small molecule produced by skeletal muscles during exercise, has beneficial effects against the hypothalamic inflammation triggered by over-nutrition. It has long been recognized that adipose tissues are a type of endocrine organ, apart from their classical role in fat storage, they also play a role in secreting a variety of soluble substances, such as pro-inflammatory cytokines and adipokines [18,19]. In particular, adipokines mediate cellular stresses, such as inflammation, oxidative stress, ER stress, and mitochondrial dysfunctions, which are directly linked to the development of metabolic disorders [20,21]. In accordance with the acceptance of the adipose tissue as an endocrine organ, a great deal of attention has been paid to the identification of novel functions of muscles, which may serve as a third type of endocrine organ. Further, it has been identified that skeletal muscles produce multiple cytokines and soluble peptides, which are regarded as myokines [22,23,24]. Since the interplay between adipokines and myokines represents a Yin-Yang model with regards to the control of body homeostasis [23], it has been well accepted that factors secreted by muscles lead to beneficial effects against multiple metabolic diseases by reversing the cellular stresses. Furthermore, exercise programs are well known to be an effective therapeutic intervention for multiple metabolic disorders [11,12,25]. Thus, it is valuable to identify the soluble substances that are secreted from skeletal muscles based on the amount of exercise or alteration of the muscle mass. Recently, irisin, a myokine that is secreted from skeletal muscles and whose levels are elevated by exercise, has been attracting attention with regards to the field of research exploring the interrelationship between muscle-derived circulating factors and their beneficial effects on human health [26,27,28]. The levels of BAIBA have also been found to be elevated in the plasma of subjects following exercise and muscle-specific PGC-1α transgenic mice [13], indicating its role in mediating the beneficial effects of exercise. Consistent with these findings, previous reports have also suggested that BAIBA plays an anti-inflammatory role in various peripheral organs that are metabolically active [15,29,30]. The hypothalamus is the central unit that controls the whole-body energy metabolism by integrating multiple afferent inputs originating from the brain and peripheral organs [1,2]. In addition, it has been well established that the disruption of this system could be a primary cause for the development of obesity [3,31]. Therefore, we investigated whether BAIBA also has reversible effects on hypothalamic inflammation, which is a primary pathogenic event associated with the perturbation of the hypothalamic circuit functions during the development of obesity. We first tested whether BAIBA affects the obesity phenotypes observed in mice fed with HFD for 20 weeks. The systemic administration of BAIBA for two months did not reverse the substantial elevations of body weight and fat mass; however, we found that BAIBA had reversible effects on the hypothalamic inflammation observed in severely obese mice fed with a fat-rich diet for a long time. These findings provide an important clue for the potential application of BAIBA for treating patients with obesity. BAIBA administration may serve as an adequate strategy for preventing obesity or inhibiting the development of obesity at the early stage. Consistent with this suggestion, we observed that BAIBA treatment during the early period of over-nutrition successfully countered the body weight gain and expansion of fat tissues resulting from the consumption of a fat-rich diet. Based on the notion that microglia, which are the resident innate immune cells in the CNS, dynamically participates in the initiation of hypothalamic inflammation during conditions of over-nutrition [3,5,6], we further confirmed that BAIBA effectively reversed the hypothalamic microglial activation in mice following both long-term and short-term consumption of a fat-rich diet. It is well-known that elevated levels of plasma-saturated fatty acids during obesity development are coupled to a variety of cellular stresses, leading to the disruption of homeostatic function and eventually causing metabolic disorders [5]. Especially, numerous studies have suggested that hypothalamic gliosis is responsible for saturated fatty acid-induced hypothalamic inflammation and the related metabolic disturbances [9]. Recently, we have also identified a marked elevation of the palmitic acid levels in the hypothalamic tissues, as well as the plasma, during the early exposure to a fat-rich diet [32]. Consistent with these evidences, we further confirmed the anti-inflammatory properties of BAIBA against palmitic acid-induced microglial activation and inflammation.

Although a majority of studies have focused on understanding the mechanisms underlying the pathogenesis of obesity and identifying strategies for treating obesity, they have been relatively relegated to providing information regarding the initiation of obesity during the early stage of over-nutrition. The current study offers persuasive scientific evidence that exercise programs serve as effective intervention strategies to suppress the initiation of obesity development, and that myokines derived from skeletal muscles can help in establishing useful preventive approaches against obesity. This study also raised an open question regarding the intracellular mechanism, which is coupled to the anti-inflammatory effects of BAIBA on hypothalamic microglia. Previous studies have indeed reported that BAIBA improves inflammatory responses in multiple cells, including vascular endothelial cells, myocytes, and adipocytes, through regulating AMP-activated protein kinase (AMPK), peroxisome proliferator-activated receptor (PPAR)-δ, and PPAR-γ signaling pathways [15,29,33]. However, further studies are needed to assess the related downstream signaling pathways and identify crucial molecular mediators associated with both the physiological and pathological contribution of circulating BAIBA in the context of obesity development linked to the hypothalamic inflammation. Collectively, our findings preliminarily suggest that BAIBA alleviates the hypothalamic inflammation by targeting the microglia, and thus, may affects the obesity pathogenesis triggered by overnutrition. These results provide novel insights into the contributions of exercise-derived endocrine factors on the hypothalamic control of energy metabolism and offers new strategies to be considered in addressing the interrelationship between metabolic abnormalities and muscle physiology.

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
