# Peer review of "Beta-Aminoisobutyric Acid Inhibits Hypothalamic Inflammation by Reversing Microglia Activation"

_cells, 2019, doi:10.3390/cells8121609_

Round 1

Reviewer 1 Report

All the requested have been answered appropriately by the authors.

Author Response

Thanks for your supports,

Jaegeun Kim

Reviewer 2 Report

I do not understand why the authors still do not indicate the mouse stereotaxic atlas that was used to locate the injection site and to identify the brain areas investigated. What is "atlas 50" (line 107)? Paxinos and Franklin's? other?

Author Response

Thanks for reviewer's the precise comment.

According to the comment, we added the information of the stereotaxic coordinates used for icv injection (line 80-81), histological experiments (line 106-109) and hypothalamic extraction for qPCR (line 135-136)  in the materials and method section of manuscript as indicated by red font. In addition, we are uploading the revised manuscript as an attached word file.

We hope that this revised manuscript will be subjected to the publication.

Thanks for your supports,

Jaegeun Kim

This manuscript is a resubmission of an earlier submission. The following is a list of the peer review reports and author responses from that submission.

Round 1

Reviewer 1 Report

Based on the fact that exercise improves the consequences of obesity, the authors suggest that some substances (BAIBA), resulting from striated muscle metabolism, could influence in that improvement. Therefore, the authors study the effects of BAIBA on hypothalamic inflammation after in vitro treatment with palmitic acid and after a feeding period with high fat diet. The results could be interesting however several points should be clarified to improve the manuscript and the understanding of the work.

No references are indicated in material and methods. The necessary references of the techniques on which they have based their study are not indicated.

2.1. The gender, total number of animals as well as the number of animals per group must be indicated.

What is the initial and final weight of the animals after the feeding period?

2.2. What stereotaxic atlas was used for icv administration and for obtaining hypothalamic samples and coronal sections?

The hypothalamus is integrated by multiple nuclei with multiple functions. Have the authors identify the nuclei in which the effects are more or less pronounced? Do they find the same effect at different levels of the hypothalamus? At what stereotaxic levels are in the images they provide in the manuscript?

In some figures the values obtained with the standard diet do not appear. Why?

What about the comparison between palmitic acid vs BAIBA in Figs 3 and 4? Some results are apparently significant.

Are there studies in which obese animals have been subjected to exercise and analyzed the cellular parameters and responses determined in the present study?

Reviewer 2 Report

Major comments:

The title does not reflect the article since BAIBA may have acted in hypothalamic inflammation and did not inhibit the development of obesity.

The description of the experimental design is confusing, without making clear the moment of the treatments. Please improve, can make scheme.

What parameter is used to define severe obesity?

Only did IL-1B Elisa, why?

What is the number of animals per group?

Why did they only make protein expression of IL1B and gene of the other proteins involved in inflammation?

 The authors cannot affirm the physiological effect based solely on gene expression.

The authors should do a test of variance and not only test t since they involve more variables.

Why in Figure 5 do the authors only consider the control group in the A/C  chart? They should standardize.

In the discussion, they could develop better based on the results of gene expression. In addition to discussing what mechanism would be acting in the reduction in mRNA.

Reviewer 3 Report

The paper by Park et al entitled ‘Beta-aminoisobutyric acid inhibits the development of obesity by reversing hypothalamic inflammation’ describes the investigation of the effect of peripheral BAIBA administration on inflammation induced in the hypothalamus in obesity induced by a high fat diet.  The authors main finding is that there is a correlation between attenuation of obesity and reduced inflammation induced by BAIBA when administered early during the weight gain period.  This correlation suggests that preventing inflammation of hypothalamus may prevent weight gain but is unable to reverse weight gain when administered at later stages of weight gain.

The paper is well written, however there are some details lacking in the materials and methods and results that make the significance of the conclusion difficult to confirm.

The number of animals used in each experiment/analysis is not clearly defined in the materials and methods and is not provided in the legends for figures 1, 4, 5.  Figure 2, refers to n=5. The reproducibility of the in vitro findings presented in in Figure 3 is also not clear.

The time-lines of the treatment is particularly important in this study and would be more clearly understood if a diagram was provided demonstrating when during the high fat diet regime and palmitic acid treatment BAIBA treatment was administered.

Results 3.1 Figure 1 clearly demonstrates that systemic BAIBA administration in drinking water after induction of obesity by high fat diet reduces peripheral (plasma) markers of inflammation but does not affect body, epididymal or liver weight.  The authors state at the end of this section that ‘These finding suggested that BAIBA-based therapies may be limited to circumstances of severe obesity and may be effective against early stage obesity development which is initiated at least m part via hypothalamic inflammation’.  This is not in fact what these findings show (although, this is the ultimate conclusion of the study).  A more appropriate conclusion for the results presented in this section is that although BAIBA administration reduces peripheral inflammation associated by a high fat diet it does not affect the obesity phenotype.  This could be because the peripheral anti-inflammatory effect of BAIBA does not reach the hypothalamus (which is the results presented in Figure 2).

Results 3.2

Figure 2A please include Iba-1 staining of control mice fed standard diet.  Also include lower power image showing the anatomical location of the image purported to be hypothalamus.  

Results 3.3

The authors explore the association between FFA activation of microglial and production of inflammatory cytokines.  Notwithstanding the lack of information regarding the reproducibility of this experiment (n not provided) this figure supports the conclusion that BAIBA preventsmicroglial activation and pro-inflammatory cytokine production.  Given the in vivo data that BAIBA attenuates inflammation after induction with HFD it would be good to see the effect of BAIBA administered after induction of inflammation by palmitic acid and conditions where BAIBA and palmitic acid were mixed prior to addition to cells to demonstrate the effect of BAIBA pre-treatment does not prevent palmitic acid activation by neutralising/  blocking the activity of palmitic acid rather than preventing/attenuating microglial activation. The specificity of this effect could be further supported by including cytokine expression that the authors predict will be not activated by palmitic acid.

Section 3.4

This section describes an experiment where the effect of systemic pre-treatmentwith BAIBA prevents microglial activation.  As per the previous section it is not clear whether BAIBA pre-treatment neutralises or blocks the activity of palmitic acid rather than preventing/attenuating microglial activation.  The title of the section and title of Figure 4 and conclusion of this section should be modified as the authors have not shown that BAIBA ameliorates activation/ reverses microglial activation because the BAIBA was administrated prior to palmitic acid treatment.

Section 3.5

This section describes the effect of BAIBA treatment after the induction of obesity phenotype and hypothalamic inflammation.  The presented results show a significant increase in some markers of metabolic activity and obesity and reduced numbers of microglial.  The authors claim that BAIBA reduced body weight gain however this result was not statistically significant (although the trend was there). This statement should be modified in the text of the results, discussion and abstract to refer to specifically refer to epididymal fat reduction or qualify the effect on body weight.  Again, this section lacks a reference to animal numbers so the basis of the statistics and the reproducibility of the effect cannot be validated.

Discussion

The authors present data that supports their conclusion that BAIBA treatment prevents or inhibits the development of obesity when administered at the early stage.  The authors appropriately do not conclude that this is directly due to attenuation of microglial activation in the hypothalamus, although the authors provide evidence to support an association between BAIBA attenuation of microglial activation and increased metabolic activity. 

Line 348 the sentence beginning ‘Collectively..’ should be modified to reflect their earlier discussion that there is an association between BAIBA induced reduction in microglial activation and obesity but that they have not shown that this is directly responsible for reduced fat and increased metabolism.

Minor

Line 16 change nature to natural

Line 52, 257, 314, 318, 327 the use of ‘reversible effects of BAIBA’ is used incorrectly

Line 205 provide reference to figure elements (D) and (E).

Line 289 delete ‘new’